# The Process of Home-Visiting Nurses Supporting People with Mental Disorders

**DOI:** 10.3390/ijerph20216965

**Published:** 2023-10-24

**Authors:** Fumi Ohtake, Maiko Noguchi-Watanabe, Kumiko Morita

**Affiliations:** Graduate School of Health Care Sciences, Tokyo Medical and Dental University, Tokyo 113-8510, Japan; noguchimaiko.chn@tmd.ac.jp (M.N.-W.); morita.phn@tmd.ac.jp (K.M.)

**Keywords:** mental disorders, home-visiting nurse, community, process, stigma, daily lives, practice, personal recovery

## Abstract

The number of people with mental disorders (PMD) living in the community is increasing; however, it is unclear how home-visiting nurses (HVNs) supporting them in the community acquire their support skills. This study aimed to reveal the process of how HVNs learn support skills for PMD. Semi-structured interviews were conducted with 14 HVNs supporting PMD living in the community. The grounded theory approach was used for data analysis. As a result, two stages were present: “Explore the personal recovery of PMD” and “Believe in the potential of PMD and accompanying them”. The first stage is further divided into two themes: “Overlapping the worlds of PMD and HVNs”, and “Easing difficulty in living for PMD”. In the first stage, HVNs gained a better understanding of PMD and obtained insight into the support they needed in their daily lives. In the second stage, HVNs became to provide the support that PMD truly needed. HVNs gained a deeper understanding of the reality of PMD through their support. After HVNs found the support PMD required, they sought to provide it, ultimately resulting in finding ways to facilitate the personal recovery of PMD.

## 1. Introduction

Since the 1950s, community transition policies have been promoted in the UK; and since the 1970s, deinstitutionalization has caused a shift to community health care [1]. This is not only in the UK but also in European countries and around the world [1,2,3]. In Japan, a community transition policy has been implemented since 2004 [4]. For over 10 years until 2016, the number of inpatients in psychiatric beds declined by 43,000 (13%) [5].

Home-visiting nurses (HVNs) are recognized as a crucial supporter of people with mental disorders (PMD) in the community. In Japan, the number of PMD of psychiatric home nursing increased from about 14,000 in 2007 to over 50,000 in 2015 [6]. The criteria for HVNs are more than 1 year of experience in supporting PMD or more than 20 h of training in psychiatric home healthcare nursing [7]. The training reinforces the knowledge base with lectures covering basic information about symptoms, medications, and assessment of mental disorders, as well as the social resources and systems available for PMD in the community, the home nursing system, and progression leading up to the current home nursing situation. The concept of psychiatric nursing is also taught in this context, including recovery, strength, and appropriate communication with patients. Following the outbreak of COVID-19, lectures are now delivered in an online format, with no opportunities for group work or other practical exercises. Accordingly, HVNs require psychiatric nursing skills derived from knowledge and experience in their practice.

There are people in the community who live with mental disorders and need care [8]. Psychiatric nursing in the community has seen situations that are complex and require immediate attention [9]. Elements of home visit nursing and intervention effects, such as a psychotherapeutic approach, have been identified in various countries [10,11,12,13]. Nevertheless, the process through which these support skills are acquired remains uncertain.

Furthermore, when supporting PMD, overall personal recovery is the ultimate goal [1]. Previously, Leamy et al. (2011) reported that an integrated concept of personal recovery was identified to have five processes: connectedness, hope and optimism about the future, identity, meaning in life, and empowerment [14]. Remarkably, the paradigm shift focuses on what PMD can do rather than on treating disease. It is influenced by the postmodernism idea that everyone is unhealthy rather than the idea of eliminating disease altogether [15]. Hence, their support is also based on positive health, emphasizing fostering resilience in PMD [16].

The content of psychiatric nursing care provided in the community has already been elucidated [17,18]; the ways in which HVNs acquire these skills remain unclear. Thus, this study aimed to clarify how HVNs conduct the process of supporting PMD in the community.

## 2. Materials and Methods

### 2.1. Study Design

We found it appropriate to use a qualitative descriptive approach to address this research question. In order to reveal the specific process of how HVNs in practice obtain nursing skills for PMD and how they provide it, grounded theory was thought to be appropriate [19]. Due to this, we used the grounded theory approach to clarify the process of HVNs’ support of PMD [20,21]. The theory was constructed from the data of the participants who experienced the phenomena [22]. We thought this method was suitable for demonstrating the process from the data.

### 2.2. Participants

This study included 14 HVNs, including 12 females and two males from three facilities. An inclusion criterion for HVNs was as follows: nurses with more than 5 years of nursing experience, regardless of home or psychiatric care. There are three reasons for setting this criterion. First, in Japan, HVNs typically work after providing inpatient care [23]. Second, most home nursing agencies provide support not only for PMD but also people with physical illness [7]. Third, a variety of mental health care services can be provided regardless of the experience [24]. For these reasons, most HVNs already have nursing experience before becoming HVNs. Hence, we thought they would have a rich vocabulary to describe their nursing experiences. An exclusion criterion included nurses who only worked with inpatient psychiatric patients, because there are differences between nursing care for inpatients and community care [17,25,26,27].

Among the participants, five were colleagues of the first author, three had met several times, and the remaining six were new acquaintances. To the best of our knowledge, all participants underwent an interview for the first time during this study. Facilities A and C were home nursing agencies that supported PMD only, whereas facility B was a home nursing agency that supported people with both mental disorders and physical illnesses. From facility A, the administrator selected eight nurses whose schedules were available, and the first author met them in person to obtain their consent for interviews. Further, at facilities B and C, the first author interacted with the eligible nurses in person, and six of the eight nurses agreed to participate in the study. Conversely, two nurses did not respond to the interview request; therefore, it was determined that their consent was not obtained. Notably, no participant who agreed to participate in the study dropped out.

### 2.3. Data Collection

The first author—a nurse with experience in supporting PMD in the community—conducted semi-structured interviews. To ensure the prevention of COVID-19 transmission, participants were informed both in writing and verbally that they could opt for online interviews in advance if they wished to do so. However, none of the participants requested to be interviewed online. Data were collected through face-to-face individual interviews. For that reason, interviews were conducted at an adequate distance between the two participants, and precautions including mask-wearing and ventilation were implemented. Questions were generated by the first author. The interview guide developed questions based on the relevant literature [28,29]. The interview guide questions were as follows: (1) Please tell us about your most memorable support experiences in supporting PMD to date. (2) What are your aims when providing support to PMD? The interview guide added that, depending on the content of the interview, there might be supplementary questions. Notably, the generated questions were kept to a minimum, in order to ensure that HVNs could speak freely about their individual support experiences [30]. The data collection period was from August to October 2020. The interviews were scheduled for 60 min during or after working hours at the participant’s place of employment. The actual interview time was 40–70 min. After being interviewed, three participants were asked about more details to define the meaning of a part of the data. This process took around 10 min. All interviews were audio recorded after obtaining the participants’ consent. Moreover, participants’ facial expressions and gestures during the interviews were added as data after the interviews. Every effort was made to ensure the participants’ comfort.

### 2.4. Data Analysis

To assess the support process, data were analyzed using the grounded theory analysis [21]. Additionally, the methods proposed by Charmaz were referred to in the integration process [20]. The audio-recorded interview data were transcribed into text. First, the first author used an initial coding based on each participant’s actions or feelings in narratives and then created a focused coding from the initial coding based on how HVNs described their feelings and support to PMD [21]. Then, the first author replaced this coding with data to make axial codes related to HVNs’ feelings and actions [20]. After analyzing all narratives, the first author compared each code and collated them with similar axial codes. Subsequently, the axial codes were integrated into a theoretical code when a new name for the code emerged [20]. This process was repeatedly checked to confirm that the theoretical code was appropriately articulated from the data. During this time, the first author discussed with colleagues if coding from the data was appropriate. Furthermore, theoretical codes were repeatedly reviewed to ensure that the content was accurately presented. Additionally, the first author noted ideas on how to proceed with the process. These theoretical codes were integrated into surmised categories to compare properties and dimensions. The categories were then polished via repeated division and integration, as well as examination, of coding properties and dimensions to clarify the concept. The first author was supervised by the co-authors to ensure that the categories that emerged were appropriate. Finally, we examined the chronological order of the individual codes and created a conceptual diagram to present the process based on the relationship among categories. Seven categories and one core category were included in the entire process. The Criteria for Reporting Qualitative Research (COREQ) checklist was applied as the reporting guideline for this study [31].

### 2.5. Rigor

The rigor of this study was determined by four criteria: credibility, transferability, dependability, and confirmability [32]. The first author learned about grounded theory in graduate lectures and also participated in grounded theory seminars and exercises several times at other universities. Two professionals supervised the analysis: one is a qualitative research expert; the other has experience in qualitative research. Furthermore, the analysis was discussed with colleagues. As this study was conducted using grounded theory and as the results could reveal processes that HVNs were unaware of, we did not present the results to the participants to ensure consistency. However, the storyline was shown to four nurses who were not involved in the interviews who were asked to confirm whether the storyline represented the process of support.

### 2.6. Ethical Considerations

The research protocol of this study was approved by the Ethics Review Committee of Tokyo Medical and Dental University School of Medicine, Japan (approval number: M2020-015). We explained the purpose of the study, its implementation, and participant rights to all the participants in writing and verbally. Furthermore, the participants signed an informed consent form. The interview was conducted in a private room where privacy could be assured, and permission was obtained to audio record the interview.

## 3. Results

### 3.1. Summary of Participants

The study participants were 14 HVNs, 12 females and two males from three facilities. The participants’ average nursing experience was 15.5 (5–29) years, with the average psychiatric nursing experience of 11.1 years (1–26 years). Furthermore, their average home-visit nursing experience was 4.5 years (5 months–18 years). Details of the same are shown in Table 1.

### 3.2. Outline of the Process

The outline reveals the process of the HVNs’ support of PMD (Figure 1). The HVNs searched for how to provide support for PMD while enhancing their understanding of PMD and the meaning of their support. The process was largely divided into two stages. During the first stage, the HVNs explored the personal recovery of PMD while supporting them. At first, the HVNs grappled with what support was to be provided to the PMD, despite having a background in the support of patients with physical diseases. That is, due to a lack of experience in being in contact with PMD and their negative perception of them, some of them were also hesitant to engage with PMD because of their negative emotions. Through direct contact with PMD, the HVNs gradually came to understand them as individuals and recognize some of their common tendencies. As they began to understand five specific characteristics of PMD, HVNs began to see areas in which the PMD were incapable of managing their lives. They provided support for these areas after understanding this concept. Additionally, they supported the PMD in stabilizing their daily lives. Nevertheless, in this stage, the HVNs could not clearly envision their personal recovery.

In the continuation of their support, the process progressed to the next stage. The HVNs came to realize that support meant facilitating the personal recovery of the PMD. They understood that personal recovery for PMD is a process in which they live their own lives with hope. For this purpose, the HVNs sought better support for the PMD, and they continued to believe in the potential of the PMD. By interacting through support, they experimentally adapted their support skills. Consequently, the HVNs’ humanity matured. They rejoiced together in the realization of the potential of the PMD.

### 3.3. Storyline for Each Stage

#### 3.3.1. Stage 1: Explore the Personal Recovery of PMD

This stage was divided into two themes: Overlapping the worlds of PMD and HVNs and Easing difficulty in living for PMD.

##### Theme 1. Overlapping the worlds of PMD and HVNs

In this theme, there are two categories: *Eliminating prejudice among PMD* and *Focusing on the difficulties in the lives of PMD*.

Some of the HVNs had never provided support to PMD. This was due to previous negative images of mental disorders and the fact that HVNs often felt that it was an obligation to visit, even though it was against their will. Some of them who chose to work in home healthcare nursing were interested in supporting the lives of people who had chronic or terminal diseases but were less interested in supporting PMD. They felt a sense of duty in visiting PMD since they had no choice but to go there. There were also negative feelings toward PMD. Some participants said that at first, they were afraid of them. They had little prior experience being involved with PMD.

Through obligatory visits and support, they noticed five characteristics of PMD (Table 2).

(1)Sensitivity related to reactions to relationships

In their interactions with the PMD, the HVNs noticed that the PMD were sensitive to interpersonal relationships. PMD can be susceptible to their environment, as in the case of one who was out of sorts after her husband started to work from home due to the pandemic, or another who was concerned about a minor relationship involvement. Occasionally, the HVNs noticed many young PMD had emotional wounds in their past, and that caused their sensitivity.

(2)Peculiar imbalance

The HVNs felt that some of the PMD had an imbalance of abilities, in that while they were highly intelligent, they seemed to be unable to care about other things, such as the abilities of housekeeping, making ends meet, and so on. For the HVNs, some of the PMD had a peculiar way of thinking; thus, they seemed to have difficulty in their living situations. The HVNs also felt that the maturity level of the PMD was slower than in other people in general. For instance, one person in his 20s behaved to his mother as if he was at a rebellious age.

(3)Poor self-esteem

The HVNs also noticed that the PMD had low self-esteem. Some of them told the HVNs that they were immature due to a lack of social experience in their lives. One seemed to be very glad being commended by their home doctor. The HVNs noticed this characteristic, because it was a different feeling that they felt themselves.

(4)Vulnerability to the challenges of survival in society

Some of the PMD had difficulty leaving positions of being supported, and the HVNs sometimes felt that the PMD were vulnerable. One who had a poor sense of social responsibility tried to work only to the limited extent that their welfare benefits were not terminated. One tried to be absent from work if they did not feel like working. Some of them were mentally dependent on their supporters. The HVNs thought the cause to be three reasons: firstly, the family background in which they grew up, seeing their parents not working and receiving welfare benefits; secondly, their lack of self-confidence due to their experiences of failure; and lastly, a situation in which they had become accustomed to being exempt from responsibility because of their disorders.

I often think, “Seriously?” when PMD for whom I have been providing support are hired and finally employed and who often call me in the morning and say, “Well, I am not feeling motivated today; should I take the day off work?” So, I told them, “Oh no, do you think I go to work fully motivated every day?”(Participant. a)

(5)Harmful effects caused by psychiatric symptoms

The HVNs realized that if psychiatric symptoms did not appear, the personalities of the PMD would come to the fore. The HVNs often found that those symptoms hid the personalities of the PMD. This would occur very quickly, and at times, the PMD seemed to enter a sensory world triggered by psychiatric symptoms; therefore, they felt that the PMD were unable to share their worsening psychological symptoms with the HVNs. Furthermore, the HVNs thought there was societal prejudice against mental disorders. One participant felt that many people who hear that a person has mental disorders generally treat the disorder as if it were a synonym with that person and react as if the person’s entire personality was lost to the mental disorder. They felt that their reactions were similar to people’s reactions to dementia.

As the HVNs began to lose their prejudice toward the PMD, they realized that there was no clear distinction between PMD and healthy people. This feeling did not disappear even when the mental symptoms of the PMD worsened and they showed crisis-related behaviors. At that point, *Eliminating prejudice among PMD* was apparent.

After that, the HVNs began *Focusing on the difficulties in the lives of PMD*. After they understood the fundamental characters of the PMD, the HVNs imagined difficulties in living based on the characteristics of PMD. However, they felt that the problem was not the characteristics themselves but that the PMD were unaware of their situation and were unable to ask for help themselves, making life difficult for them.

After her condition deteriorated, we could not talk about such things (e.g., dinner menus and other ordinary conversations) anymore. Looking back at it now, she said something contrasting, like “I am doing fine,” when I asked her if she was having any problems. She would not listen to me anymore (when her mental condition deteriorated). I used to be able to have a conversation with her. Although (since her mental condition has deteriorated) she interrupted me when I asked her something that she did not want me to ask.(Participant. k)

##### Theme 2. Easing difficulty in living for PMD

In this theme, there are three categories: *Struggling to support using firsthand nursing skill and intuition*, *Maintaining a balance between the world of PMD and the living environment*, and *Encouraging PMD to accept themselves*.

At first, the HVNs found a support purpose of making daily living for the PMD easier. For that, the HVNs began *Struggling to support using firsthand nursing skill and intuition* without necessarily being aware of the purpose of the support to the personal recovery of the PMD. The HVNs conducted assessments in order to support the physical health of the PMD at first, because they felt unsure of the assessment for mental disorders. However, the HVNs assessed the mental condition of the PMD from their activities of daily living, such as separating garbage properly. The HVNs understood the signs of their worsened mental condition even without direct inquiry, such as when the PMD neglected to perform tasks they used to do routinely. Furthermore, the HVNs assessed the capacity of the PMD to perform daily life tasks. This assessment extended beyond routine tasks like keeping oneself clean and household budgeting, as the HVNs also considered the actions of the PMD within specific contexts, such as assisting their families in hosting relatives during traditional Japanese events like Obon. Additionally, the HVNs understood the PMD in a sensible way. For example, some of the HVNs felt that it was a sign of trust that the PMD allowed them to enter into their homes.

The HVNs learned through their practice. They gained a slightly better understanding of PMD. The HVNs acquired knowledge from their own experiences and other HVNs’ practices. In addition, they deepened their understanding of the mental state by asking other PMD who had similar experiences and by asking doctors. Many participants expressed their experience of acquiring knowledge naturally in practice rather than actively acquiring knowledge through participation in lectures or training. For example, the HVNs used communication skills they had gained. It was not training, but they obtained these skills from the reactions of the PMD they cared for.

Based on that, the HVNs endorsed routines in the daily life of the PMD to make life comfortable, such as cleaning rooms, household financial managing, and exercising. They felt that these forms of support led to an overall improvement in their mental conditions and an advancement in their lives, including a desire to work. Moreover, they were aware that they were on an equal footing with the PMD. The HVNs found that the power of their position was not as strong in the community as it was in the hospitals. The HVNs then understood that they could interact with the PMD as individuals, without being overly conscious of their role as nurses. The HVNs adjusted to the standards and pace of the PMD. They adopted a person-centered approach by customizing their support to align with the needs of the PMD. In cases where the PMD might not have admitted that their mental condition worsened, the HVNs respected their standards, even though assessments indicated that their mental condition worsened. Their consideration involved building a relationship of trust, and they created intuitively tailored support for the needs of the PMD, such as fulfilled the required role for PMD. They played a range of roles from friend and family to counselor. They learned to infer their needs.

Therefore, participants sometimes found it difficult to explain their forms of support in words. They supported the PMD through trial-and-error learning from their practices. They considered the timing of the support by predicting the reactions of the PMD. However, there were times when the HVNs received unexpected reactions from them.

When I said (to him), “You depend on your mother, right?”. I tried to rouse him but he said, “Yes, you’re right. I’m spoiled. So be it”. […] He took a so-what attitude. I thought he react something that if I tell him like that (but he did not).(Participant. a)

Secondly, the HVNs tried *Maintaining a balance between the world of PMD and the living environment*. Sometimes, the PMD had subjective perspectives that stemmed from their characters or psychiatric symptoms like delusion. At the time, the HVNs developed support to prioritize maintaining the community life of the PMD. The HVNs accepted the worlds of the PMD without denial. The HVNs worked with the PMD to understand how to address the issue of approaching their symptoms from a life perspective. This was because the aim of the HVNs’ support was to enable the PMD to continue living in their communities. When the PMD suffered from auditory hallucinations or delusion, the HVNs diverted their attention from their mental symptoms by, for instance, listening to the favorite music of the person in their care. In addition, they investigated delusions in a practical way. When the individual suspected that someone stole their cell phone information, the HVN went with him or her to a cell phone store to verify the facts. This decreased the pain from their symptoms and sometimes helped them avoid trouble in their communities.

I could not say it was delusion because it was difficult for him to understand what was delusional. So, in a way, I would slip through with him.

I told him that he would be arrested if he tried to attack his neighbor, and I asked him if he had suffered any damage, and he said no. So, I said, “Well, let us wait and see,” or “Let me know if there’s anything else,” something like that.(Participant. g)

Additionally, they assisted the PMD in developing social skills, such as forming relationships with others in society. In most cases, the closest relationships for the PMD were with their families. The HVNs aimed to create good relationships with the PMD and their families. For that, they assessed the needs of the family members of the PMD as well. In addition to this, some of them mediated between the PMD and their bosses to facilitate comprehension of what was expected from their companies. The HVNs supported the situation from an objective standpoint. They thought that it would help broaden the perspectives of the PMD. However, when the HVNs explained what the companies expected from the PMD, they chose their words carefully in order not to hurt them.

Furthermore, the HVNs collaborated with other supporters like doctors, public health nurses, nurses in hospitals, city hall staff, and welfare supporters, because there were various resources in the community that the PMD could take advantage of.

Lastly, the HVNs were *Encouraging PMD to accept themselves*. They empathized with the thoughts and feelings of the PMD after excluding their assumptions and prejudices. For instance, the HVNs could not understand how they felt about their psychiatric symptoms themselves, but they could empathize with the pain caused by them. The HVNs’ empathy helped them in promoting self-acceptance among the PMD. Sometimes, the PMD had guilty feelings when they could not do something, and they blamed themselves. The HVNs provided positive meaning to the experiences of the PMD. As a result, the HVNs enhanced their self-esteem. On the other hand, they did not always empathize with the need for escape felt by some of the PMD when they felt like escaping to not confront their actual situations or escaping a challenge to change.

She said she was not able to go (to the welfare facility for the disabled). […] I was like, you were able to take a rest, right? Then, she made her face like this (that she did not understand the meaning), so I said, “You might say I was able to take a rest; it is fine.” So, her face turned like this (she was surprised at the difference in meaning).(Participant. m)

By providing such support, the HVNs stood on the side of the feelings of the PMD, seeing that these attitudes had a positive impact on them. That led them to make trust the foundation of the support. Trust was created not only via direct support to the PMD but also from necessity, such as in the case of cleaning one’s old father when it was discovered that stool had been leaking from his pants. The HVNs felt that intervening in the difficulties of the PMD was more likely to lead them to feel that the support was useful and helpful, which in turn was more likely to lead to the next level of support.

#### 3.3.2. Stage 2: Believe in the Potential of PMD and Accompanying Them

In this stage, there are two categories: *Bringing PMD close to the lives they want* and *Resonating with PMD*.

The HVNs integrated their capabilities for assisting the PMD close to them in living their own way. Moreover, the HVNs’ feelings were changed through supporting them. Their motivation increased after they were grateful for the changes in the PMD.

The HVNs offered support for *Bringing PMD close to the lives they want*. They provided support as the PMD wished. They noticed that supporting what the PMD desired to do in their lives was the most important support. Occasionally, it appeared to be difficult to find, or it appeared as though there was nothing to support for them; this resulted in frustration for the HVNs. However, they detected that maintaining their lives in their way was the purpose for supporting if the PMD rejected receiving support for bath taking or room cleaning. Because they appeared to feel fine without those forms of support. As a result, they provided meaning to the support. The HVNs were conscious of increased life choices among the PMD. It meant not only some concrete examples like participating in community groups or obtaining a new job; sometimes, they felt that the mental growth of the PMD led to increase their life choices.

It was not easy, and they struggled to find ways to support them. Thus, they assisted the PMD step by step and enhanced their capacity for living. Although participating in society was the ultimate goal, some PMD were still in a situation where they were unable to manage their household chores at that time. Hence, the HVN set their initial support to the purpose of washing used dishes with them. After that, seeing the response of the PMD to the support, they gradually expanded their forms of support. For instance, they proposed going to a welfare facility for the disabled since those activities were connected to society.

Furthermore, the HVNs executed far-sighted support for the future of the PMD. For example, they thought some of the PMD would require capabilities following the passing of their caregivers, such as an elderly mother. In preparation for the future, they began visiting their home to establish a relationship.

Conversely, sometimes the HVNs believed there was no way to intervene in negative situations where the PMD did not accept their reality and tried to escape it. At the time, one of the HVNs felt conflicted about being unable to talk about their negative feelings to the PMD or their colleagues. They felt some of the PMD did not wish to be on their own because of their dependence on supporters. However, they could not say that to the PMD directly. Moreover, one participant felt that the PMD had a choice to not accept support if they did not want to. Nevertheless, this person could not express their feelings to their colleagues about this, because the other HVNs seemed to not have the choice to withhold support.

As they supported the PMD, the HVNs began *Resonating with PMD*. They could see the viewpoints of the PMD after they understood their realities, except in the cases when the PMD avoided facing their harsh reality. Moreover, the HVNs came to notice slight growth in the PMD. Continuing to offer support, the HVNs tried to maintain their motivation. In order to accomplish this, the HVNs noted positive actions or reactions from the PMD. For example, one began to wash their face before the HVN visit; another began to describe one’s feelings before venting their discontentment. The ability to realize slight growth in the PMD and reflect it in their support was enriched as they became more experienced in providing support. This led to a greater variety of support from the HVNs.

They truly felt they could support the lives of the PMD in the community, not in hospitals. It became a pleasure to provide support that was unique to the community. By contrast, when the PMD did not exhibit a desire to improve their lives or the time came to end support because the PMD began to support themselves, some of the HVNs felt the pain of termination of support.

Finally, the HVNs matured emotionally and mentally. They realized their own growth when they felt the growth of the PMD, and it affected their own self-fulfillment. As individuals, the HVNs developed through their support. Through the process, the theme *Support the personal recovery of PMD* was revealed as a core category for the HVNs.

I believe it is all gone if I abandon him. Right now, I think my challenge is how far I can trust and accompany him. Well, it is always easy for me to give up and dismiss him, but how should I say… I impose myself to trust his power and work with him without doing that. (So, what are you doing for that?) Well, I treasure the moment when I see his action as something shiny. Not shiny, but for me, it might look shiny. For example, he ran all his errands, such as he followed a procedure at the city hall and saw a doctor the next Monday while money was scarce.(Participant. a)

## 4. Discussion

This study clarified the process of HVNs supporting PMD living in the community. Based on the results, we will discuss the following three standpoints.

### 4.1. Beginning to Support PMD without Prejudice

Many participants experienced anxiety and did not know what to do for the PMD at first contact. Notably, this stigma is a widespread phenomenon in society [33]. It is also found among nurses toward PMD [34,35,36,37,38,39]. Especially in Japan, it was affected by the previous unjust segregation policy and the Japanese name for schizophrenia before the 2002 revision. This policy compelled PMD to be long period inpatients. Furthermore, the previous Japanese translation of schizophrenia was “mind-split-disease”, and the meaning of schizophrenia was defined as “the disease of disorganized mind”. [40]. Hence, the stigma was perpetuated, as it was difficult to imagine the recovery of PMD since people did not feel familiar with them [35,41,42,43]. Accordingly, the HVNs might have had negative emotions resulting from this stigma. Regarding this, reflective practice would be one effective method for confronting this [44]. Reflection aid them to deepen their introspection. It results in HVNs first realizing their own stigma.

Furthermore, the practice contributed to not only eliminating stigma, but also noticing the characteristics of the PMD. Based on this, they were able to provide effective support for the PMD. Therefore, HVNs need to learn the individual characteristics of the PMD in their clinical practice, which then leads to an understanding of the PMD. Before nursing PMD in the community, HVNs who had no experience in supporting them needed to receive lectures, gaining knowledge of supporting PMD in the community. Additionally, the content of the lectures not only included nursing methods, but also dedicated much attention to the knowledge of social systems, social resources, treatments, and medications relevant to PMD. From this study, it is suggested that HVNs may benefit from role-playing or other exercises aside from lectures that would allow them to learn by interacting with others. Such activities could improve their understanding of communication skills and their ability to seek support.

Moreover, some of the HVNs had a few opportunities to contact the PMD daily. Therefore, opportunities for performing interactions with the PMD and learning their narratives could be effective for stigma reduction [38,45]. Recovery colleges are education-based centers where individuals, family members, and supporters can learn together about recovery from mental and physical conditions [46,47,48,49]. In Japan, there are several recovery colleges as well [50]. It would be beneficial to promote the understanding that PMD are on equal footing and to reduce stigmas. In conclusion, it is necessary for HVNs to understand them individually and experientially.

### 4.2. Making the Daily Lives of PMD More Comfortable

The participants gradually approached the PMD with existing nursing skills and began learning more about PMD. This was a cycle that deepened both their support skills and knowledge. From this, it is considered that the HVNs developed their nursing skills as their understanding grew. At first, they applied their skills that they already had from previous supporting experience. Furthermore, they used assessment skills from the live environment of the PMD. Nurses were required to make comprehensive assessments of individuals [51]. It is especially expected in the community because the lifestyles of PMD are totally different. Thus, these assessment skills are needed for HVNs. In addition, they were unsure if their own assistance was suitable. Thus, they consulted and learned from their colleagues to ascertain the usefulness of their support. It led to better practices in teaching each other. They did not take much time to learn in daily practice. Accordingly, it is considered good practice to communicate well and share skills with colleagues.

Moreover, the participants overcame their stigma and realized that PMD also had self-stigma that narrowed their choices for seeking a better life. Hence, they used psychiatric nursing techniques that encouraged the PMD to accept themselves in order to reduce their self-stigma. A previous study reported that care increased recovery orientation [52,53]. They gained new psychiatric skills like communication skills. These skills are required and must also be shared in their daily practice. Additionally, participants used their own characteristics. Some of them emphasized the strengths of the PMD and allowed them to gain confidence; others gave them an objective opinion when the PMD mentioned a negative side of themselves. Therefore, the HVNs needed to be aware of the mental effects of their characteristics. Furthermore, HVNs had little opportunity to experience how other colleagues provide care because they generally visited the PMD alone. Thus, they needed to consult with their colleagues and share information about care. A colleague’s unique response and advice can be considered as an idea, or new ideas of support can emerge from it.

In addition to support, when the mental health of an individual worsened, participants thought that there were only a few things they could do to provide support. However, in addition to direct support, the HVNs needed indirect, more diverse, and complex support through collaboration with multiple professional groups, including approaches for the environment of PMD [54]. Although the HVNs were not specialized in the knowledge of crisis intervention, their support role involved coordinating with other professionals to optimize interventions [55,56]. By realizing that multidisciplinary coordination is a form of support, HVNs may be able to intervene consciously, as they are one of supporters, and increase their job satisfaction. Although the HVNs encountered difficulties in supporting the PMD, overcoming some of these difficulties was not always a problem for the PMD. In such cases, it is also important to respect their freedom of choice to receive support. For recovery, it is necessary to prioritize their decision [15]. Furthermore, the HVNs realized that the most important thing in the support of PMD is that they could continue to live in the community. For this reason, they attempted to make adjustments so that the PMD could continue to live in the community despite their psychiatric symptoms. This was a specific purpose for the HVNs.

We found that the HVNs did not utilize all skills from the beginning but acquired them in the process of support. The HVNs were expected to seek improved support for the PMD and continually consider effective support. The development of skills is believed to have been mastered in the process.

### 4.3. Attaining the Deepening of Emotions through Supporting the Entire Life of the PMD

By knowing the PMD, the HVNs were motivated to support them in their daily life. Sometimes, the HVNs found great joy in discovering their imperceptible social growth. Personal recovery has multiple aspects to its meaning, ranging from process to progress [15,57]. Understanding personal recovery helped the HVNs support them. They understood this meaning from their practice. This led to an increase in their motivation for the personal recovery of the PMD. Nevertheless, supporting PMD in the community can sometimes take years. For this, evaluating their changes can help HVNs in continuing support. Furthermore, their joy not only changed their motivation to support but also enriched their sensitivity. The feeling of sympathy had a positive impact on the personal recovery of the PMD [58]. Thus, their personal recovery benefited both the PMD and HVNs.

Conversely, the HVNs had negative feelings toward the PMD who found no hope when they could not provide the support that would lead to recovery. Furthermore, they found it challenging to discuss these emotions with their colleagues, as they were afraid of their feelings being rebuffed by other HVNs. It became clear that the HVNs needed to recognize the need to share their thoughts and emotions with their colleagues. Emotional coping was also significant for the HVNs [59]. It also helped prevent their demotivation to provide support.

Finally, HVNs’ satisfaction with their work as nurses could improve the quality of their nursing care. Many home nursing agencies are privately owned and operated owing to the fact that such an agency is one of the businesses that nurses can start in Japan. Although increasing the frequency of visits to such agencies can directly increase profits [60], prioritizing profits can lead to decreased motivation and burnout in HVNs [61]. Feeling that work is worth doing can therefore ensure the quality of nursing care and maintain the motivation of HVNs. This would also contribute to the management of nursing agencies. Developing an understanding of the personal recovery of PMD facilitates motivation to support them and contributes to one’s own growth as a human individual.

This study revealed the process by which HVNs improved their support to PMD living in the community. Compared with previous research, the process of acquiring the elements of psychiatric nursing in the community by HVNs through practice was evident. This study will provide a guide for HVNs to reflect on the deepening of their skills and may help in the educating of psychiatric nursing. They obtained skills step by step through their practice. The support itself does not change significantly in the process. However, the HVNs’ understanding of the meaning of support for PMD changed dramatically. They came to understand it was for their personal recovery. This led the HVNs to use their own sensitivities to offer support. This was a more meaningful support for the PMD, as well as a support for the HVNs, to mature in their humanity. Regardless of their career, the HVNs’ commitment to maximizing the benefits of the PMD created personalized and original support. However, as they built their careers, they gained more depth in their insights into support. The insights gained through experience were acquired as wisdom that was most helpful in supporting individual HVNs and were applied in their daily support. From this, it was clear that the most necessary skill for them was to acquire a sense of the support that PMD truly need through their practice. Furthermore, at times, this sensory acquisition was not conscious to the HVNs. Therefore, it also became clear that this skill itself should not be conscious as an objective. The effort to find better support for PMD will ultimately lead to the acquisition of skills for HVNs. Therefore, HVNs must always need to think about what support will make life better for PMD in front of them, which will be a way of acquiring the skills. HVNs can check this process against their own practice to see where they are in their practice. It may also help to provide evidence in psychiatric nursing education that mastering skills requires not only the learning of skills but also the accumulation of experience.

### 4.4. Strengths and Limitation

The process by which HVNs support PMD living in the community is useful for reflecting on the support steps that HVNs are experiencing and for having a clear perspective on the purpose of the support. However, as this research focused on nursing for PMD, we did not provide the status of their recovery stage. Additionally, we did not conduct an interview from their perspective. Thus, further studies are warranted to gain more detailed insights by clarifying the support specific to their situation and conducting interviews with PMD.

## 5. Conclusions

This study revealed the process of HVNs support to PMD in the community. First of all, it began by the HVNs reducing their own unconscious stigma towards the PMD. Simultaneously, it was also about the HVNs becoming aware of their characteristics, which led to an understanding of how to support them. Next, the HVNs focused on the difficulties that PMD experience in their daily lives due to their characteristics and began to support them in alleviating these difficulties. To do this, the HVNs used existing nursing skills and new psychiatric nursing skills they had acquired while practicing support to the PMD; HVNs’ nursing practices aimed at alleviating difficulties PMD had in their lives and enabling them to continue living in the community. Finally, the HVNs shifted their support to bring the PMD closer to the life they desired. In their support, the HVNs themselves came to resonate with the joy of the growth of PMD together. These forms of support facilitated their personal recovery. This contributed to the HVNs’ own growth as well.

## Figures and Tables

**Figure 1 ijerph-20-06965-f001:**
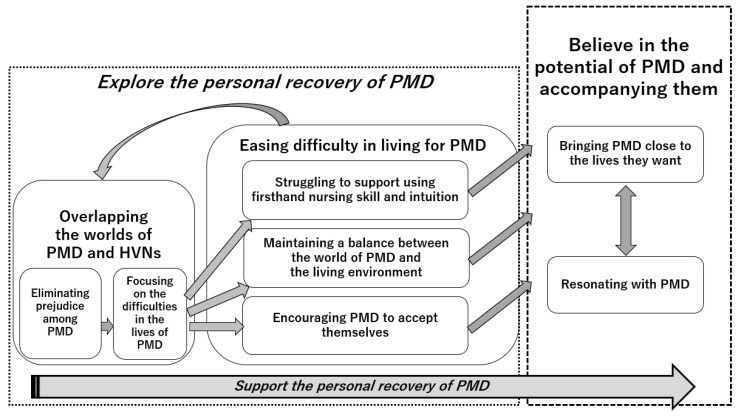
The process by which the HVNs supported the personal recovery of the PMD. PMD = people with mental disorders; HVNs = home visit nurses.

**Table 1 ijerph-20-06965-t001:** Participant Summary. (*n* = 14).

Item	Classification		Average (SD)
Gender	Male	2	
	Female	12	
Visiting nurse station	Facility A	8	
	Facility B	5	
	Facility C	1	
Age		28–58 years	43.4 (8.6)
Education level	Graduate school	3	
	University	5	
	Vocational school	6	
Years of nursing experience		5–29 years	15.5 (6.8)
Years of psychiatric nursing experience		1–26 years	11.1 (8.5)
Years of visiting psychiatric nursing experience		5 months–18 years	4.5 (5.6)

**Table 2 ijerph-20-06965-t002:** Five characteristics the HVNs noticed about PMD.

(1) Sensitivity related to reactions to relationships (2) Peculiar imbalance (3) Poor self-esteem (4) Vulnerability to the challenges of survival in society (5) Harmful effects caused by psychiatric symptoms

PMD: people with mental disorders; HVNs: home visit nurses.

## Data Availability

Not applicable.

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
