# Peer review of "The Process of Home-Visiting Nurses Supporting People with Mental Disorders"

_ijerph, 2023, doi:10.3390/ijerph20216965_

Round 1
Reviewer 1 Report
Comments and Suggestions for Authors
Thank you for the opportunity to review your research. Your research is very interesting and important.
Introduction
What are the international challenges in caring for people with mental illness in the community?
How will this paper contribute internationally?
Methods
What were the items in the interview guide?
What did you refer to when creating the interview guide?
How did the authors share or collaborate in the data analysis?
Please explain your strategy for ensuring rigour, citing the literature.
Results
Did participants with five months of experience have sufficient experience to talk about the research topic?
Discussion
What were the new findings of your research compared to previous research?
It is better to consider the potential for international contribution rather than to describe something specific to Japan.
Author Response
Thank you for your valuable comments. Your suggestions have helped us improve our manuscript. We have made revisions based on your comments, as explained below.
Introduction
1 What are the international challenges in caring for people with mental illness in the community?
→We appreciate your comment. There are people in the community who live with mental disorders and need care. Home health care in the community can be complex and require quick action. Although the individual roles of psychiatric home nursing, psychoeducation, and other elements of psychiatric home nursing have been identified in many countries, it is not clear how these elements are acquired as useful skills for practice, which we considered to be an international challenge. We are adding this to the introduction.
P1-2 lines 42-46
There are people in the community who live with mental disorders and need care [8]. Psychiatric nursing in the community has seen situations that are complex and require immediate attention [9]. Elements of home visit nursing, intervention effects such as psychotherapeutic approach, have been identified in various countries [10-13]. Nevertheless, the process through which these support skills are acquired remains uncertain.
2 How will this paper contribute internationally?
→Thank you for your valuable feedback. We are adding to the discussion that this study will provide a guide for home visit nurses to reflect on the deepening of their skills and may help in the educating of psychiatric nursing. We are adding this to the discussion.
P12 lines 556-559
Compared with previous research, the process of acquiring the elements of psychiatric nursing in the community by HVNs through practice was evident. This study will provide a guide for home health nurses to reflect on the deepening of their skills and may help in the educating of psychiatric nursing.
P12 lines 575-578
HVNs can check this process against their own practice to see where they are at in their practice. It may also help to provide evidence in psychiatric nursing education that mastering skills requires not only the learning of skills, but also the accumulation of experience.
Methods
3 What were the items in the interview guide?
→Thank you for your question. We are adding 2 items of the interview guide to the data collection.
P3 lines 102-106
The interview guide questions were: (1) Please tell us about your most memorable support experiences in helping PMD to date. (2) What are your aims when providing support to PMD? The interview guide added that, de-pending on the content of the interview, noted that there may be supplementary questions.
4 What did you refer to when creating the interview guide?
→We referred two relevant literature. We are adding that to the data collection.
P3 lines 101-102
The interview guide developed questions based on the relevant literature [28,29].
5 How did the authors share or collaborate in the data analysis?
→Thank you for your question. We are adding that how to collaborate in this study to the data analysis.
P3 lines 127-128
During this time, the first author discussed with colleagues whether coding from the data was appropriate.
P3 lines 134-135
The first author was supervised by co-authors to ensure that the categories that emerged were appropriate.
6 Please explain your strategy for ensuring rigour, citing the literature.
→Thank you for your query. We are adding that to the rigour.
P3 lines 142-143
The rigor of this study was determined by four criteria: credibility, transferability, de-pendability and confirmability [32].
Results
7 Did participants with five months of experience have sufficient experience to talk about the research topic?
→Thank you for your insightful comment. This participant has seven years of experience as a nurse, including five years in pediatrics and at least one year in a psychiatric hospital. Prior to one’s experience as a home visit nurse, one had experience caring for people with mental disorders and others, so we thought one had enough experiences to tell the topic of this research. And the participant talked about experiences that became an important part of the results, such as the support that PMD with psychiatric symptoms to be able to continue living in the community. Moreover, one provided rich data that broadened our perspective, like the unique characteristics of home visit nursing in comparison to one’s previous nursing experiences in hospitals. For that, we believe one had sufficient experience.
Discussion
8 What were the new findings of your research compared to previous research?
It is better to consider the potential for international contribution rather than to describe something specific to Japan.
→Thank you for your suggestion from a wide perspective. We found that there is a process acquiring the psychiatric nursing skills in home visit nursing by nurses. Previous research had identified the elements of the process, but not how it is acquired. It may also help to provide evidence in psychiatric nursing education that mastering skills requires not only the learning of skills, but also the accumulation of experience. The clarification of the process by which nurses acquire psychiatric nursing skills is considered a finding that will contribute to nursing education not only in Japan but also internationally. This is a duplicate of the previous answer, but we are adding that to the discussion.
P12 lines 556-559
Compared with previous research, the process of acquiring the elements of psychiatric nursing in the community by HVNs through practice was evident. This study will provide a guide for home health nurses to reflect on the deepening of their skills and may help in the educating of psychiatric nursing.
P12 lines 575-578
HVNs can check this process against their own practice to see where they are at in their practice. It may also help to provide evidence in psychiatric nursing education that mastering skills requires not only the learning of skills, but also the accumulation of experience.
Reviewer 2 Report
Comments and Suggestions for Authors
Very interesting and important topic for nursing educators in the field of visiting nursing education.
I would like to help improve the quality of the paper by providing the following feedback.
1. Introduction : I believe that the research trends on the topic and the reason why grounded theory analysis is necessary should be described.
2. Materials and Methods
2.3. Data collection : You should write down a list of questions you used in the interview.
2.5. Rigor : In order to secure the rigor of the research, it is necessary to provide information on the preparation process for researchers to write grounded theory ( participation in research or classes,, etc.).
3.3. Storyline for Each Stage
-Please remove the underline from the italicized sentences.
4. Discussion
You have described the discussion in detail. I would like to include a part about nursing career. Research participants have diverse careers. It is judged that support for PMD is different between HVNs with 5 months’ career and those with over 20 years. Although it is a qualitative study, it would be nice to show the differences in their approaches. I hope junior nurses know that the ‘weight of years’ called ‘career’ can develop nursing capabilities called ‘wisdom’.
This work inspires my own interest in nursing education research! Thank you!
Author Response
Thank you for your valuable comments. Your suggestions have helped us improve our manuscript. We have made revisions based on your comments, as explained below.
- Introduction : I believe that the research trends on the topic and the reason why grounded theory analysis is necessary should be described.
→We appreciate your comment. We are adding what was found by previous research and what is unknown. Also, we have added a simple sentence to the introduction because we thought the reason why grounded theory analysis we selected is already described in the study design.
P1-2 lines 42-46
There are people in the community who live with mental disorders and need care [8]. Psychiatric nursing in the community has seen situations that are complex and require immediate attention [9]. Elements of home visit nursing, intervention effects such as psychotherapeutic approach, have been identified in various countries [10-13]. Nevertheless, the process through which these support skills are acquired remains uncertain.
P2 lines 62-64
In order to reveal the specific process how HVNs in practice obtain nursing skills for PMD and how provide it, grounded theory thought it would be appropriate [19].
- Materials and Methods
2.3. Data collection : You should write down a list of questions you used in the interview.
→Thank you for your suggestion. We are adding that to the data collection.
P3 lines 102-106
The interview guide questions were: (1) Please tell us about your most memorable support experiences in helping PMD to date. (2) What are your aims when providing support to PMD? The interview guide added that, depending on the content of the interview, noted that there may be supplementary questions.
2.5. Rigor : In order to secure the rigor of the research, it is necessary to provide information on the preparation process for researchers to write grounded theory ( participation in research or classes,, etc.).
→Thank you for your suggestions. We are adding that to the rigor.
P3 lines 143-145
The first author learnt about grounded theory in graduate lectures, also participated grounded theory seminars and exercises several times at other universities.
3.3. Storyline for Each Stage
-Please remove the underline from the italicized sentences.
→Thank you for your advice. We are removing that.
- Discussion
You have described the discussion in detail. I would like to include a part about nursing career. Research participants have diverse careers. It is judged that support for PMD is different between HVNs with 5 months’ career and those with over 20 years. Although it is a qualitative study, it would be nice to show the differences in their approaches. I hope junior nurses know that the ‘weight of years’ called ‘career’ can develop nursing capabilities called ‘wisdom’.
→Thank you for your great suggestion. We are adding description regarding career and wisdom to the result and the discussion.
P10 lines 426-429
The ability to realize slight growth in PMD and reflect it in their support was enriched as they became more experienced in providing support. This led to a greater variety of support by HVNs.
P12 lines 564-568
Regardless of their career, the HVNs' commitment to maximizing the benefits of PMD created personalized and original support. However, as they built their careers, they gained more depth in their insights into support. The insights gained through experience were acquired as wisdom that was most helpful in supporting individual HVNs, and were ap-plied in their daily support.
Round 2
Reviewer 2 Report
Comments and Suggestions for Authors
Thank you for your efforts.